# Radiomics Based on Nomogram Predict Pelvic Lymphnode Metastasis in Early-Stage Cervical Cancer

**DOI:** 10.3390/diagnostics12102446

**Published:** 2022-10-10

**Authors:** Xueming Xia, Dongdong Li, Wei Du, Yu Wang, Shihong Nie, Qiaoyue Tan, Qiheng Gou

**Affiliations:** 1Department of Head and Neck Oncology, Cancer Center, West China Hospital, Sichuan University, Chengdu 610041, China; 2Department of Radiotherapy, Cancer Center, West China Hospital, Sichuan University, Chengdu 610041, China; 3Department of Network Engineering, School of Computer Science and Engineering, South China University of Technology, Guangzhou 510641, China; 4Department of Radiology, The Affiliated Hospital of Southwest Medical University, Luzhou 402103, China

**Keywords:** cervical cancer, pelvic lymph node metastasis, radiomics, nomogram

## Abstract

The accurate prediction of the status of PLNM preoperatively plays a key role in treatment strategy decisions in early-stage cervical cancer. The aim of this study was to develop and validate a radiomics-based nomogram for the preoperative prediction of pelvic lymph node metastatic status in early-stage cervical cancer. One hundred fifty patients were enrolled in this study. Radiomics features were extracted from T2-weighted MRI imaging (T2WI). Based on the selected features, a support vector machine (SVM) algorithm was used to build the radiomics signature. The radiomics-based nomogram was developed incorporating radiomics signature and clinical risk factors. In the training cohort (AUC = 0.925, accuracy = 81.6%, sensitivity = 70.3%, and specificity = 92.0%) and the testing cohort (AUC = 0.839, accuracy = 74.2%, sensitivity = 65.7%, and specificity = 82.8%), clinical models that combine stromal invasion depth, FIGO stage, and MTD perform poorly. The combined model had the highest AUC in the training cohort (AUC = 0.988, accuracy = 95.9%, sensitivity = 92.0%, and specificity = 100.0%) and the testing cohort (AUC = 0.922, accuracy = 87.1%, sensitivity = 85.7%, and specificity = 88.6%) when compared to the radiomics and clinical models. The study may provide valuable guidance for clinical physicians regarding the treatment strategies for early-stage cervical cancer patients.

## 1. Introduction

Cervical cancer is the fourth most common malignant tumor and the fourth leading cause of cancer death in women worldwide [1]. Due to the promotion of tumor screening, an increasing number of early cervical cancers have been discovered in recent years [2,3]. In the European Society for Medical Oncology (ESMO) Clinical Practice Guidelines on cervical cancer, radical hysterectomy with bilateral lymph node dissection [with or without sentinel lymph node (SLN)], carried out by laparotomy, was regarded as standard treatment in patients with the International Federation of Gynecology and Obstetrics (FIGO) stage IA2, IB and IIA [4,5]. However, nearly 30% of early-stage cervical cancer patients have pelvic lymph node metastasis (PLNM), which is an extremely important factor affecting treatment decisions [6,7]. For cervical cancer patients with PLNM, concurrent chemoradiation is preferred instead of surgical treatment, according to ESMO Clinical Practice Guidelines [8]. A considerable number of patients have received unnecessary radical hysterectomies, which may lead to serious complications and reduce their quality of life [4,5]. In addition, PLNM is a poor prognostic indicator for recurrence and metastasis in early-stage cervical cancer [9,10]. Therefore, the accurate prediction of the status of PLNM preoperatively plays a key role in treatment strategy decisions.

Magnetic resonance imaging (MRI) has been widely recommended as the optimal imaging equipment for preoperative staging and lymph node exploration of cervical cancer [11]. However, traditional MRI based on morphological assessment, such as lymph node size and morphology, has only a low sensitivity of 75% for lymph node status [12]. Due to the presence of inflammatory hyperplasia and micrometastatic lymph nodes, the efficacy of traditional MRI in differentiating lymph node status is difficult to satisfy [13]. Low diagnostic efficiency may result in many cervical cancer patients being understaged, which affects clinical decision-making [14].

Radiomics, which can extract quantitative features from digital medical images and convert them into mineable high-dimensional data, plays an important role in personalized clinical decision-making [15]. Radiomics analysis can be used in the diagnosis, prognosis, and prediction of curative effects in a variety of tumor types by developing appropriate model refinement features. This method has shown a huge potential for predicting lymph node metastatic status in a wide range of tumor types in a cost-effective and non-invasive way [16,17,18]. T2-weighted MRI imaging has also been widely utilized in the staging of cervical cancer [19].

Therefore, the purpose of this research was to develop and validate a radiomics-based nomogram for the preoperative prediction of pelvic lymph node metastatic status in early cervical cancer.

## 2. Materials and Methods

### 2.1. Patient

This retrospective study was approved by the hospital ethics committee, and the requirement for informed consent was waived. We retrospectively collected patients diagnosed with cervical cancer with biopsy-proven cervical carcinoma receiving initial treatment with surgery in our hospital from February 2017 to October 2019. The inclusion criteria were as follows: (i) patients receiving radical hysterectomy and pelvic lymphadenectomy; (ii) pathologically diagnosed cervical cancer; (iii) pretreatment MRI scan available in our hospital; (iv) primary cancer lesions visible on sagittal T2WI; and (v) available clinical characteristics. The exclusion criteria were as follows: (i) patients receiving other treatment before surgery, including neoadjuvant chemotherapy, radiotherapy, or conization; (ii) absence of preoperative MRI scan in our hospital; (iii) poor MRI image quality, for example, indistinct MRI image of the cervical structure; and (iv) rare pathological types of cervical tumor, for example, mucoepidermoid carcinoma [20]. The MRI images were reviewed by two radiologists with 7 and 9 years of experience.

In total, 150 patients were enrolled. Among these patients, 35 patients with PLNM and 115 patients without PLNM were treated surgically and confirmed pathologically. Baseline clinicopathologic characteristics, including age, FIGO stage, maximal tumor diameter (MTD), histological subtype, and stromal invasion, were derived from medical records. The patient selection process is shown in Figure 1.

### 2.2. MRI Acquisition Protocol and Tumor Segmentation

The MRI scans were performed on each patient before the initial surgical treatment. A T MRI scanner was used (Achieva 3.0 T, Philips, Amsterdam, The Netherlands), which was equipped with a 16-channel abdominal coil. Abdomen and pelvic MRI examinations were performed on patients. To avoid the loss of image information, the Digital Imaging and Communications in Medicine (DICOM) images from Picture Archiving were acquired without any compression or downsampling. The conventional protocol included T2-weighted imaging on the axial, sagittal, and coronal planes, T1-weighted imaging, fat-suppressed T1-weighted imaging, fat-suppressed T2-weighted imaging, and diffusion-weighted imaging (b = 0.800 s/mm^2^) on the axial planes. The scanning parameters for the fat-suppressed turbo spin echo (TSE) T2-weighted images were as follows: TR/TE: 4854/85 ms, FOV = 300 × 300 mm; matrix = 232 × 171; slice thickness/gap: 5/1 mm; NEX = 2.

All of the MRI images were loaded into 3D slicer software and then manually three-dimensionally segmented (open-source software; https://download.slicer.org/, accessed on 1 January 2022). All of the manual segmentations of the primary tumor region were contoured as the region of interest (ROI) on axial T2WI, and the coronal and sagittal MRI images were used for guiding the segmentation of the ROIs in the cross-sectional plane. Segmentation was performed by a radiologist who had 7 years of experience in gynecological MR imaging and resolved any uncertainty through consultation with another radiologist who had 9 years of experience. The representative images of the lesions are shown in Figure 2. The workflow of the radiomics analysis is presented in Figure 3.

### 2.3. Radiomic Feature Extraction

Following the hand segmentation of the tumor, the radiographic characteristics of the tumor were extracted using open-source software called Pyradiomics. To generate a consistent normal-distribution image distribution, standardized techniques on T2WI pictures were applied. We collected 1688 radiological characteristics from seven imaging types on the T2WI of the tumor. Tumor size (e.g., volume), shape (e.g., circumference, diameter), grayscale cooccurrence matrices (e.g., energy, contrast, entropy), grayscale run-length matrices, and grayscale dependency matrices are all quantified using these characteristics. Python 3.7 is used to implement all functionalities. The workflow of the radiomic feature extraction is presented in Figure 3.

### 2.4. Radiomics Feature Selection and Development of the Radiomics Model

The goal of feature selection is to choose some of the most useful features from the original features to minimize dimensionality and increase the model’s generalization ability, as well as to speed up the fitting process. We have chosen two types of feature selection methods: filtered and embedded. MINE, Pearson’s initial relation number, and ANOVA F value (F Classif) are chosen in the filtered scheme, whereas ExtraTree, GBDT, and Random Forest are chosen in the embedded scheme.

Because we used Pyradiomics to extract a vast number of features from the images, these features are not fully associated with the prevalence of the disease. Therefore, we chose the most effective feature extraction approach among six alternative feature filtering strategies to implement feature extraction. We also evaluated the validation of Chalkidou et al. [21], in which research showed that a stable model might be built using 10–15 characteristics. The outcomes of the experiments reveal that not all datasets will benefit from such a selection.

We use a support vector machine model to implement the sample classification when we have finished selecting features. The RBF kernel is also chosen in the SVM model, and the hyperparameter C is chosen to optimize the AUC value under the training set.

To develop the clinical prediction model, the stromal invasion depth, MTD, and FIGO phases were examined. To determine the sample classes, the SVM model is employed. The logistic regression approach with forward stepwise selection was used to create the combined model. In the combined model, the radiomics signature and clinical risk variables were incorporated. The combined model was presented as a radiomics-based nomogram to make it a more user-friendly tool for the preoperative prediction of PLMN status. The formula of the radiomics signature of the final radiomics model is shown in Appendix A.

### 2.5. Assessment of Predictive Models

The performance of these prediction models was evaluated in the training cohort before being verified in the validation cohort using the receiver operating characteristic (ROC) curve. The agreement between the nomogram prediction probabilities of the PLMN status and actual results was assessed using a calibration curve.

### 2.6. Statistical Analysis

All of the statistical analyses were conducted with R 3.4.1 and Python 3.7. The independent-sample t-test of independent samples was used to assess the significance of age and MTD between the training cohort and the validation cohort. The chi-squared test or Fisher’s exact test was used to evaluate the significance of categorical variables such as FIGO stage, histology type, stromal invasion, lymphovascular invasion (LVSI), and nerve invasion (NI) between the training and validation cohorts. Two-tailed *p*-values less than 0.05 were considered statistically significant.

## 3. Results

### 3.1. Patients’ Clinicopathologic Characteristics

Between February 2017 and October 2019, a total of 301 patients who underwent surgery for cervical cancer were enrolled. According to the inclusion and exclusion criteria, 151 patients were excluded. Finally, 150 patients fulfilled the eligibility criteria and were enrolled in the following analysis (Figure 1).

The patient characteristics are summarized in Table 1. The distribution of clinical characteristics (age, FIGO stage, MTD) and pathological characteristics (histology type, stromal invasion depth, LVSI, nerve invasion) were balanced between the training and validation cohorts. The MTD, stromal invasion depth, and LVSI status showed a significant difference between the patients with and without PLNM metastasis in the training cohorts. The LVSI status and NI status showed a significant difference in the validation cohort, as shown in Table 1.

### 3.2. Feature Selection and Performance of the Clinical Model and Radiomics Model

We collected a total of 1688 characteristics, with 744 associated with wavelets, 372 associated with LBP, 186 associated with squares, and 107 associated with the original. The exponential, the gradient, and the logarithmic functions each have 93 characteristics. Furthermore, feature filtering was accomplished utilizing two types of methods: filtered and embedded. Overall, the filtered technique beats the embedded extraction strategy, ignoring the fact that the embedded scheme varies significantly with different feature extraction ratios.

Although the AUC increases by over 90% as the number of features embedded grows, the outcomes are more heavily influenced by the number of chosen features. In contrast, when the number of features grows, the filtered solution maintains a consistent rising trend. MINE outperforms F Classification and Pearson in regard to feature selection, with a rank of 138. The MINE approach is focused on a feature number of 138 for later trials (Figure 4).

In the training cohort (AUC = 0.925, accuracy = 81.6%, sensitivity = 70.3%, and specificity = 92.0%) and the testing cohort (AUC = 0.839, accuracy = 74.2%, sensitivity = 65.7%, and specificity = 82.8%), the clinical models that combined stromal invasion depth, FIGO stage, and MTD performed poorly (Table 2). In both the training (AUC = 0.975, accuracy = 91.8 percent, sensitivity = 92.0 percent, and specificity = 91.7 percent) and testing cohorts (AUC = 0.852, accuracy = 77.1 percent, sensitivity = 82.9 percent, and specificity = 71.4 percent), the radiomics model outperformed the clinical model significantly, as depicted in the ROC curves (Figure 5).

### 3.3. Performance of the Combined Model and the Radiomics Nomogram

Tumor stage, tumor infiltration depth, and radiomics signature were chosen throughout the creation of the integrated model. The combined model had the highest AUC in the training cohort (AUC = 0.988, accuracy = 95.9%, sensitivity = 92.0%, and specificity = 100.0%) and the testing cohort (AUC = 0.922, accuracy = 87.1%, sensitivity = 85.7%, and specificity = 88.6%) when compared to the radiomics and clinical models (Table 2 and Figure 5). The combined model has a better performance than the clinical model in both the training and test cohorts. The radiomics-based nomogram and the combined model are shown in Figure 6. The calibration curves of the radiomics-based nomogram demonstrated satisfactory agreement between the predictive and observational possibility of the PLMN status in both the training and validation cohorts.

## 4. Discussion

For patients with early-stage cervical cancer, pelvic lymph node (LN) status is one of the most important factors taken into account when making clinical decisions regarding surgery or radical chemoradiotherapy. Many inspection methods have been made to enhance the accurate evaluation of LN status before surgery, such as sentinel lymph node (SLN) biopsy, magnetic resonance imaging (MRI), and positron emission tomography (PET)-computed tomography (CT). SLN biopsy can acquire optimal diagnostic efficiency, but the absence of a standard surgical technique and the surgeon’s training and experience may influence the outcome of SLN biopsy [22]. In addition, SLN biopsy is an invasive approach and is not routinely applied. MRI has widely been used in preoperative staging evaluation and lymph node detection of cervical cancer. Although MRI has the potential to identify LNM status according to the MRI morphological appearances of PLNM, such as size and shape, its efficiency and sensitivity in diagnosing PLNM are unsatisfactory [23]. Several previous studies have shown that a considerable proportion of cervical cancer patients were misclassified according to morphologic criteria on MRI images [24,25]. Although PET-CT shows favorable performance and is superior to MRI, considering the availability of PET-CT devices and high inspection costs, its wide application has a long way to go [26].

In this study, we successfully developed and validated a noninvasive individualized radiomics-based nomogram integrating the radiomics signature and clinicopathologic factors for predicting PLNM in patients with early-stage cervical cancer before surgery. The proposed radiomics nomogram exhibited a favorable performance in discriminating lymph node status in the training and validation sets. This radiomics nomogram also showed more predictive efficacy than the traditional diagnostic criteria of PLNM.

Chalkidou and colleagues proposed a solution to the problem of picking a limited number of finite features [21], which is the focus of this study. The results of the studies reveal that only a subset of datasets fit the criteria, not all of them. As part of our related work on lymphovascular space invasion, we substituted the study population in the original dataset in this experiment. It was not possible to achieve a decent result in Lasso’s technique based on the theory presented by Chalkidou et al. To achieve this, two broad categories of feature selection strategies were chosen for validation, and all experiments consistently indicated that even the same dataset might have a significant influence on the selected features when the experimental aim is changed.

The findings of the experiments indicate that the tree-based feature extraction approach may not be suited for feature extraction in imagingomics. The tree scheme is not ideal for processing continuous features, and the continuity of the final derived features is quite visible. Second, the tree-based algorithm’s performance tends to be poor when processing data with substantial feature correlation, and the generated features are highly correlated. The other set of approaches, on the other hand, makes better use of data correlation and hence beats the tree-based strategy in the experimental comparison. MINE is the most effective of these approaches since it finds not only the linear but also the nonlinear correlations between the variables. It is possible that this is why the MINE scheme outperforms the others.

By combining the radiomics signature with easily available, preoperative clinical risk factors, we developed a quantified radiomics nomogram, which is convenient to utilize for clinicians. The radiomics nomogram could generate a personalized probability of PLNM for patients before surgery, which could provide more information for clinicians to make treatment decisions. This individualized treatment is in line with the trend of personalized precision medicine [27].

In our study, some clinical characteristics were independently correlated with LNM status, including MRI-reported LN status and FIGO stage. These clinical characteristics were quite similar to those from other previous studies based on conventional MRI analytical methods, indicating that the LNM status was closely related to the total status of the primary tumor [24,25,28]. Recently, some studies [29,30,31] developed combined models for predicting LNM status in cervical cancer patients based on clinical and histological information and MRI images. These studies utilizing the radiomics method showed comparable prediction performance with ours. Compared with these previous studies, our study provides an easier visual tool for doctors to evaluate the PLNM status preoperatively and noninvasively.

There were several limitations in our study. First, we performed radiomics analysis on the T2WI sequence. During the segmentations, we excluded the T1WI for the unsatisfactory performance in displaying the lesions. Therefore, we did not continue the segmentations on T1WI. We also excluded DWI due to the low signal-to-noise ratio and motion artifacts. In future research, we will combine or compare more MRI sequences to improve diagnostic efficiency. Second, the cervical lesions that were difficult to recognize or invisible on MRI were excluded, which may lead to potential selection biases. Third, due to the relatively small number of cases in this single-center retrospective study, a larger sample size, multiple centers, and prospective datasets are needed to optimize the performance of the radiomics model in the future. Finally, all of the manual segmentations of the tumor tissues on axial T2WI and contrast-enhanced T1-weighted spin–echo images were performed by a radiologist who had 5 years of experience in gynecological MR imaging, and each segmentation was validated by a senior radiologist who had 12 years of experience. Therefore, the final segmentations were revised and verified by two experienced radiologists. However, the concordance between the two observers was not measured. In the future, we will perform the analysis of concordance to provide more precise data. Automatic tumor segmentation and extraction with machine learning could be further explored [32].

## 5. Conclusions

This study provides an effective and noninvasive tool for the individualized preoperative prediction of PLNM status. This radiomics-based nomogram would aid the selection of the optimal therapeutic strategy and clinical decision-making for individuals. The study may provide valuable guidance for clinical physicians regarding treatment strategies for early-stage cervical cancer patients.

## Figures and Tables

**Figure 1 diagnostics-12-02446-f001:**
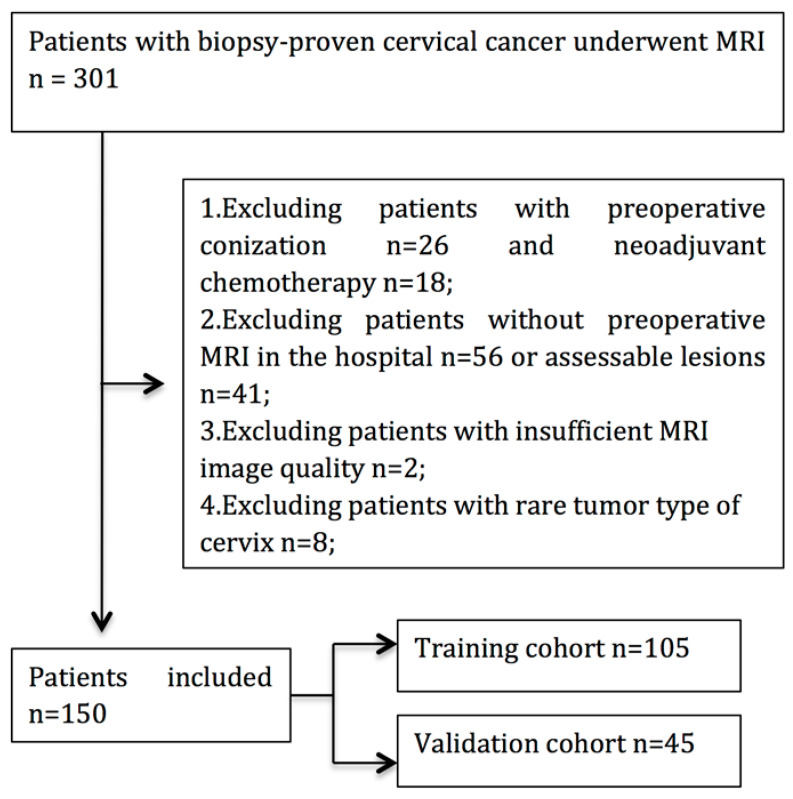
Flow chart of patient enrollment in this study.

**Figure 2 diagnostics-12-02446-f002:**
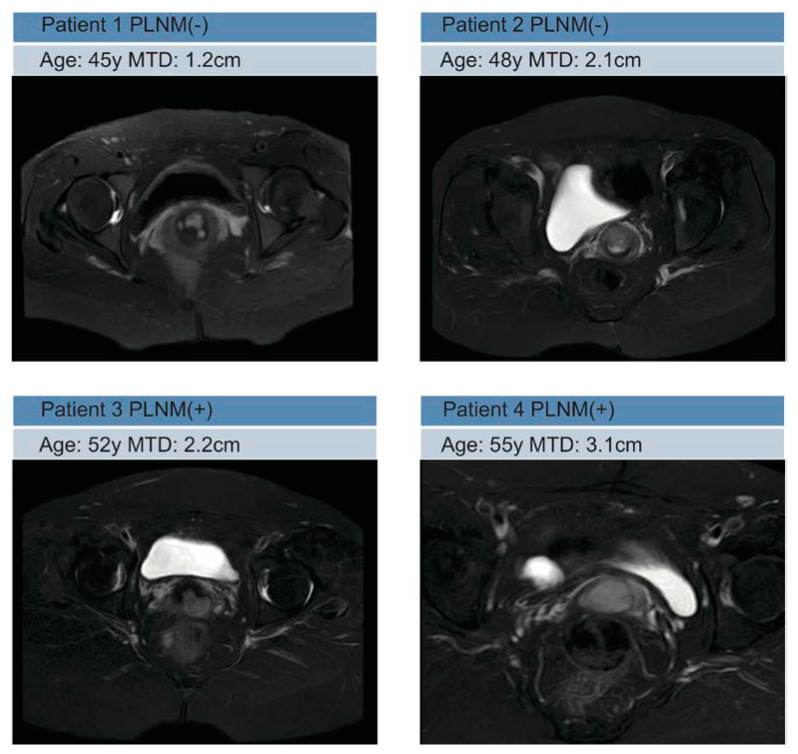
Representative MRI images in PLMN(−) and PLMN(+) patients.

**Figure 3 diagnostics-12-02446-f003:**
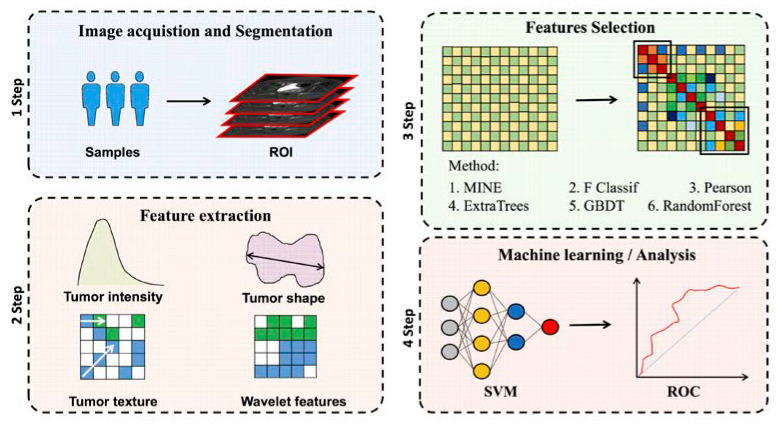
Radiomics analysis workflow. The study’s subject underwent T2WI image collection. Radiomics performed feature extraction after manually marking the ROI region of the tumor lesion. The efficient feature combination was filtered using one of two feature extraction methods: filtered or embedded. For further statistical analysis, ROC curves were used.

**Figure 4 diagnostics-12-02446-f004:**
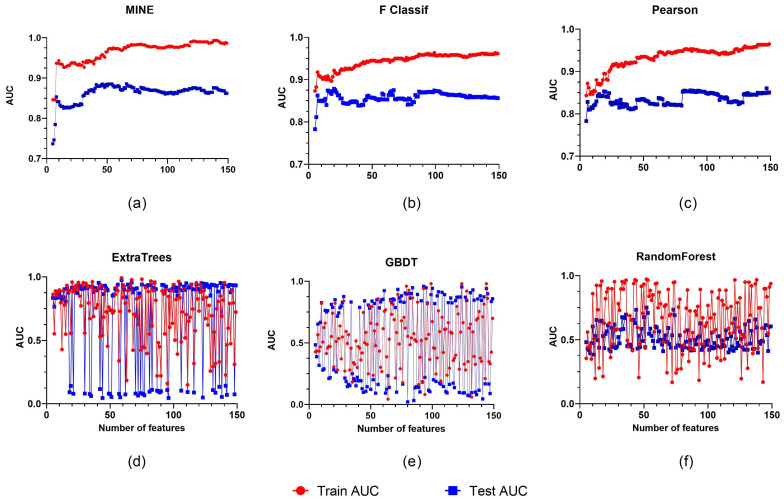
Six different feature screening approaches are illustrated, with the horizontal coordinate showing the number of features selected and the vertical coordinate showing the classifier’s AUC. The filtered scheme includes (**a**–**c**). The embedded scheme includes (**d**–**f**).

**Figure 5 diagnostics-12-02446-f005:**
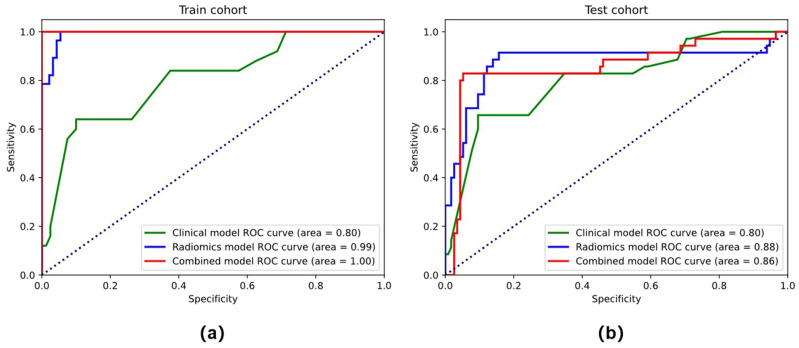
The ROC curves of the clinical model, radiomics models, and combined model in the training cohort (**a**) and testing cohort (**b**).

**Figure 6 diagnostics-12-02446-f006:**
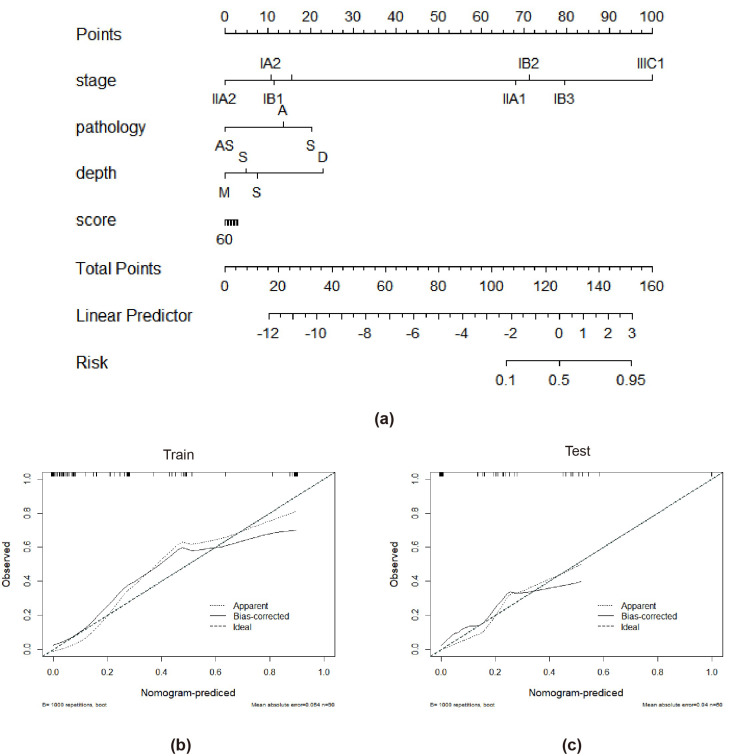
The radiomics-based nomogram (**a**). Calibration curves in the training cohort (**b**) and validation cohort (**c**). A closer fit to the diagonal line indicates a better evaluation.

**Table 1 diagnostics-12-02446-t001:** Characteristics of the included patients.

Characteristics	Training Cohort (*n* = 104)			Validation Cohort (*n* = 45)			*p* *
	PLNM(+)	PLNM(−)		PLNM(+)	PLNM(−)		
	(*n* = 25)	(*n* = 80)	*p*	(*n* = 10)	(*n* = 35)	*p*	
Age, years			0.095			0.312	0.520
Mean	47.12	46.66		43.30	46.60		
FIGO Stage (N, %)			0.787			0.208	0.871
IA	0	2		1	0		
IB	15	52		5	22		
IIA	10	26		4	13		
MTD, cm			0.007			0.220	0.815
Mean	3.95	2.77		2.97	3.07		
Histology (N, %)			0.331			0.657	0.664
SCC	19	67		10	30		
AC	4	11		0	4		
ASC	2	2		0	1		
Stromal Invasion			<0.001			0.261	0.448
Deep 1/3	21	28		6	10		
Middle 1/3	2	24		2	12		
Superficial 1/3	2	28		2	13		
LVSI status			<0.001			0.010	0.519
Positive	2	57		1	21		
Negative	23	23		9	14		
NI status			0.627			0.016	0.185
Positive	23	76		6	33		
Negative	2	4		4	2		

Note: *p* is derived from the chi-squared test or Fisher’s exact test between patients with and without PLNM in the training and validation cohorts, respectively. *p* * represents the difference in each clinicopathological variable between the training and validation cohorts. Abbreviations: PLNM: pelvic lymph node metastasis; MTD: maximal tumor diameter; LVSI: lymphovascular invasion; SCC: squamous cell carcinoma; AC: adenocarcinoma; ASC: adenosquamous carcinoma; NI: nerve invasion.

**Table 2 diagnostics-12-02446-t002:** Performance of models.

		Training Cohort				Testing Cohort			
		AUC	ACC (%)	SEN (%)	SPE (%)	AUC	ACC (%)	SEN (%)	SPE (%)
Clinical	stage	0.832	0.816	0.708	0.920	0.767	0.771	0.657	0.886
	pathology	0.476	0.531	0.08	0.96	0.527	0.485	0.08	0.886
	diameter	0.712	0.714	0.75	0.68	0.728	0.729	0.771	0.685
	Stage+pathology+diameter	0.925	0.816	0.703	0.920	0.839	0.742	0.657	0.828
Radiomics model		0.975	0.918	0.920	0.917	0.852	0.771	0.829	0.714
Combined model		0.988	0.959	0.920	1.00	0.922	0.871	0.857	0.886

## Data Availability

Raw data can be shared from the first author if there is a reasonable request.

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
