# Peer review of "Radiomics Based on Nomogram Predict Pelvic Lymphnode Metastasis in Early-Stage Cervical Cancer"

_diagnostics, 2022, doi:10.3390/diagnostics12102446_

Round 1
Reviewer 1 Report
Dear Authors, thanks for the paper about a relevant topic. The identification of lymphnodes metastasis prior surgery it's relevant for sure.
Just some questions and specifications:
- I would clarify in the text the full MRI scan protocol
- Why did you chose to do segmentations and radiomic features extraction only in T2w imaging?
- Did you consider also DWI? (I guess it was excluded for the low resolution)
- It is not clear how many features were selected. Please clarify.
- Did you consider to perform segmentations by two independent observers to see the concordance? (rather than one radiologist only?)
We may proceed further after the answers to the questions above.
Best Regards
Reviewer 2 Report
Rapid technological progress in medicine gives us the tools to help make decisions. One of them is radiomics. The decision as to whether or not to perform a lymphadenectomy in cervical cancer, and to what extent, will become more and more difficult as we think more often about not doing it. The multifactorial nature of this decision makes it necessary to use a system that takes all these factors into account. The article shows that, following the example of breast cancer, one can build a decision system that works and can be another card on the decision table from which we have to get up together with the patient with the best possible, well-balanced decision.
I have minor remarks only.
Abstract is clearly written. I would add MRI to the sentence "Radiomics features were extracted from T2 weighted imaging (T2WI)"
MAIN TEXT
Minor remarks:
1. Page 1, lines 44-46: Radiotherapy is not a standard option for FIGO stage I CC. Ref. 4 is in outdated. Please use an other citation. Whenyou write on oncological treatment it would be of value to give the ESMO guidelines: https://www.esmo.org/guidelines/guidelines-by-topic/gynaecological-cancers/cervical-cancer/eupdate-cervical-cancer-treatment-recommendations. There is stated: "In the ESMO Clinical Practice Guidelines on cervical cancer, radical hysterectomy with bilateral lymph node dissection [with or without sentinel lymph node (SLN)], carried out either by laparotomy or laparoscopy, was regarded as standard treatment in patients with FIGO (Fédération Internationale de Gynécologie et d’Obstétrique) stage IA2, IB and IIA, if the patient does not wish to preserve fertility."
2. Page 2; Citation no 8 is not sufficient to support the statement on lines 49-51.
3. Page 2; lines 51-53: I guess it goes by lymphadenectomy but it is unclear.
4. I think that giving one reference to support this finding is not enough in the context of the current study (lines 52-54).
5. Page 2; "T2 weighted MRI imaging has also widely utilized in the staging of cervical cancer." (Lines 71-72) - I suggest adding a reference (maybe one of those already provided).
Materials and Methods
6. Page 3, lines 87-88: What does it mean "poor MRI image quality"? (please give an example) and "rare pathological types of cervical tumor"? (for example: other than squamous cell carcinoma, adenocarcinoma and adenosquamous carcinoma).
7. Page 4, line 100: ... were performed (not "conducted on patients"). It is obvious that it goes by patients.
8. Page 5, lines 133-138: this text should be regular and you should refer the reader to the Figure 3.
9. Page 6, line 170: the abbreviation "NI" (first time in the text) is not explained
References:
In general, there is a lack of journal name in every entry.
9. Please check this reference for accuracy.
